# Development and Evaluation of Gelatin-Based Gummy Jellies Enriched with Oregano Oil: Impact on Functional Properties and Controlled Release

**DOI:** 10.3390/foods14030479

**Published:** 2025-02-02

**Authors:** Mariana Ganea, Potra Cicalau Georgiana Ioana, Timea Claudia Ghitea, Liana Ștefan, Florina Groza, Olimpia Daniela Frent, Csaba Nagy, Claudiu Sorin Iova, Andrada Florina Schwarz-Madar, Gabriela Ciavoi, Laura Gratiela Vicas, Pelea Diana Constanta, Corina Moisa

**Affiliations:** 1Pharmacy Department, Faculty of Medicine and Pharmacy, University of Oradea, 1st December Square 10, 410073 Oradea, Romania; mganea@uoradea.ro (M.G.); daniela.olimpia@yahoo.com (O.D.F.); lvicas@uoradea.ro (L.G.V.); corinamoisa@hotmail.com (C.M.); 2Department of Dental Medicine, Faculty of Medicine and Pharmacy, University of Oradea, 1st Decembrie Street, 410073 Oradea, Romania; cicalau.georgiana@uoradea.ro (P.C.G.I.); gciavoi@uoradea.ro (G.C.); 3Department of Obstetrics and Gynecology, Faculty of Medicine and Pharmacy, University of Oradea, 1st December Square 10, 410073 Oradea, Romania; lianaantal@gmail.com (L.Ș.); schwarz_andrada@yahoo.com (A.F.S.-M.); diana_pelea@yahoo.com (P.D.C.); 4Department of Preclinics, Faculty of Medicine and Pharmacy, University of Oradea, 410068 Oradea, Romania; florinamiere@uoradea.ro (F.G.); soriniova03@gmail.com (C.S.I.); 5Doctoral School of Biomedical Science, University of Oradea, No. 1 University Street, 410087 Oradea, Romania; nagycsaba95@yahoo.com

**Keywords:** functional foods, gelatin-based jellies, oregano oil, controlled release, swelling index, disintegration time, tensile strength, natural extracts, bioactive compounds, food formulation

## Abstract

Functional foods play a crucial role in contemporary dietary strategies. This study investigates the incorporation of oregano oil, a bioactive extract that is known for its antimicrobial and antioxidant properties, into gelatin-based gummy jellies to develop functional food products with controlled release properties. The jellies were evaluated for mass uniformity, swelling index, disintegration time, and tensile strength under simulated oral and gastric conditions. The results showed that oregano oil significantly reduced the swelling index (e.g., 128.76 ± 0.67% at pH 5) and prolonged the disintegration time (e.g., 6–18 min across pH environments), highlighting its potential for controlled release. The mechanical strength remained stable (5.2 ± 0.3 N), ensuring structural integrity. These findings suggest that oregano-oil-enriched gummy jellies offer health benefits, although further studies are needed to explore their long-term stability and bioavailability.

## 1. Introduction

In recent years, there has been an increasing interest in functional foods, which are designed to provide health benefits beyond basic nutrition [1]. These foods often contain biologically active components such as vitamins, minerals, probiotics, or plant-based compounds that have positive effects on health beyond mere nutrition [2]. Functional foods are an integral part of modern dietary strategies aimed at improving overall health, preventing chronic diseases, and enhancing the quality of life [3]. As consumer awareness grows, there is a demand for more natural and health-promoting ingredients in food products, driving innovations in the formulation of functional food items [4].

Gummy jellies have become one of the most popular delivery systems for functional ingredients. These gelatin-based confections have transformed from simple candies into effective carriers of health-promoting compounds like vitamins, antioxidants, and plant extracts [5]. The appeal of gummy jellies lies in their ease of consumption, palatability, and ability to encapsulate various functional ingredients in a matrix that maintains stability and effectiveness over time [6]. This versatility makes gummy jellies ideal for incorporating natural extracts, which are often sought after for their potential health benefits [7].

Natural extracts, particularly those derived from herbs, fruits, and essential oils, are rich in bioactive compounds that have been shown to possess antioxidant, anti-inflammatory, and antimicrobial properties [8,9,10,11]. Incorporating these extracts into functional foods, such as jellies, offers an innovative way to enhance their nutritional profile. For example, plant-derived extracts like oregano oil, rich in phenolic compounds such as carvacrol and thymol, have been recognized for their strong antimicrobial and antioxidant activities [12,13], as have other plants such as *Artemisia annua* [14]. These natural extracts not only improve the health benefits of the food product but can also act as natural preservatives, extending the shelf life of the product without the need for synthetic additives [15].

In this context, gummy jellies have emerged as an attractive medium for delivering these bioactive compounds, offering a convenient, enjoyable, and effective way for consumers to integrate health-promoting ingredients into their daily diets [16]. However, the incorporation of natural extracts into jellies poses certain challenges, including potential interactions between the gelatin matrix and the bioactive compounds, which can influence the texture, stability, and release properties of the final product [17]. Additionally, factors such as the pH of the medium, the solubility of the extracts, and the conditions of processing and storage play a significant role in determining the functionality and effectiveness of the natural ingredients [18].

Gelatin was chosen as the matrix for incorporating oregano oil due to its unique properties that make it highly suitable for controlled release applications. Derived from collagen, gelatin forms a robust three-dimensional network that is capable of encapsulating bioactive compounds while maintaining mechanical stability. Its compatibility with hydrophobic substances like oregano oil ensures a uniform distribution and retention within the matrix [19]. While alternatives such as agar and pectin offer plant-based solutions and are increasingly popular for their vegetarian and allergen-free characteristics, they lack the same degree of gel strength and flexibility that are provided by gelatin. These factors, combined with gelatin’s proven track record in pharmaceutical and food applications, justified its selection for this study. Future research may explore plant-based polymers as alternative matrices to address sustainability and dietary preferences [6,9,20,21].

This study evaluates gelatin’s potential as a carrier for oregano oil in functional jellies, balancing delivery efficiency with sustainability and allergen concerns. Future research could explore plant-based alternatives to meet ethical and dietary needs.

## 2. Materials and Methods

### 2.1. Equipment

An ultrasonic water bath; silicone molds; sterile gloves; a refrigerator; an analytical balance, Kern ABT 220-5DNM (Kern and Sohn GmbH, Balingen, Germany); watch glass; a spatula; 10 mL Berzelius beakers; a 100 mL graduated cylinder; a volumetric flask with a volume of 1000 mL; Electrolab TDT-08L apparatus (USP); a tester; and a pH meter, the wireless HALO_2_ (Hanna Instruments, Cluj-Napoca, Romania) were used.

### 2.2. Reagents

Oregano oil (Doterra, Pleasant Grove, UT, USA); Grape Oil (Madonna di Lugo, Costa D’Oro, Italy); date syrup (Bio Planet, Leszno, Poland); citric acid (Decorias, Rediu, Romania); gelatin (Merck, St. Louis, MO, USA); glycerin (Farmachim 10 SRL, Ploiești, Romania); distilled water (Decorias, Rediu, Romania); NaCl—Sodium Chloride (Decorias, Rediu, Romania); HCI—Concentrated Hydrochloric Acid (Sigma Aldrich, Taufkirchen, Germany); KCI—Potassium Chloride (Sigma-Aldrich, Taufkirchen, Germany); CaCI_2_ × 6 H_2_O—Chloride calcium dehydrate (Sigma-Aldrich, Taufkirchen, Germany); NaH_2_PO_4_ × H_2_O—sodium diacid phosphate hydrate (Sigma-Aldrich, Taufkirchen, Germany); Na_2_HPO_4_—disodium acid phosphate (Sigma-Aldrich, Taufkirchen, Germany); KH_2_PO_4_—potassium diacid phosphate (Sigma-Aldrich, Taufkirchen, Germany); NaHCO_3_—sodium acid carbonate (Decorias, Rediu, Romania); Na_2_SO_4_—sodium sulfate (Sigma-Aldrich, Taufkirchen, Germany); NH_4_CI—ammonium chloride (Sigma-Aldrich, Taufkirchen, Germany); KSCN—sodium thiocyanide potassium (Sigma-Aldrich, Taufkirchen, Germany); Urea (Sigma-Aldrich, Taufkirchen, Germany); and Na_2_S × 9 H_2_O—sodium sulfide nona-hydrat (Sigma-Aldrich, Taufkirchen, Germany) were used.

### 2.3. Preparation of JO/EJO Gummy Jellies

The preparation of the gummy jellies with/without essential oil of oregano (JO/EJO) was carried out according to the methods described by Vojvodić Cebin, A., et al. and Matulyte, I., et al., with some modifications [16,19]. For the preparation of the gummy jellies without oregano oil, it was necessary to prepare the gelling agent, namely the 20% gelatin solution.

The gelatin solution was obtained by hydrating it for 10 min in a mixture of glycerin and distilled water (1:4 ratio) at room temperature, and then, it was placed in an ultrasonic water bath at a temperature of 75 °C to dissolve for 15 min. After that, at the same temperature and using a magnetic stirrer with a hot plate, the following were added to the container containing the gelling agent: 0.5 g of 1% citric acid solution, 20 g of date syrup, 5 g of grape seed oil, and the rest of the distilled water while carefully stirring (1000 rpm). This was to promote the homogeneous dispersion in the gelatin mass of all components while at the same time avoiding excessive aeration. In the case of the gummy jellies based on oregano oil, the procedure was exactly the same, with the difference that 0.5 g of oregano essential oil was added to the formula at the end under continuous stirring (500 rpm) after the gelatinous mass had cooled 10 °C.

The addition of grape seed oil serves multiple purposes in the formulation. Firstly, it acts as a carrier oil, enhancing the dispersion of oregano oil within the gelatin matrix, ensuring a uniform distribution of the active compound. Secondly, grape seed oil is rich in antioxidants, which may contribute to the overall functional properties of the gummy jellies, complementing the antioxidant potential of oregano oil. Lastly, it helps improve the texture and mouthfeel of the jellies, contributing to their sensory acceptability. These combined roles make grape seed oil a vital component of the formulation.

All the materials that were used in the preparation and the quantities in which they were used are shown in Table 1.

The resulting gummy jelly mass was carefully poured into bear-shaped silicone molds using a plastic Pasteur pipette with a large opening and then allowed to cool for 30 min at room temperature. After cooling, they were removed from the molds using sterile gloves, individually packaged in plastic containers, and then packed in transparent polypropylene zip-lock bags and kept in a refrigerator (at a temperature of 4 °C) until further analysis.

The method of preparation of the gummy jellies is shown schematically in Figure 1.

### 2.4. Determination of Mass Uniformity and OJ Yield

The analysis of the uniformity of the mass of the jellies that were loaded with oil of oregano (JO) was carried out according to the method of determining the uniformity of the mass, described in the *Romanian Pharmacopoeia 10th Ed.*, in the Compressi monograph. Tabulettae [22], and according to the study by Hassen Elshafee, A., and Moataz El-Dahmy, R. [23], to which some changes were made. To carry out this test, the average mass of twenty OJs was calculated after weighing them individually on the Kern ABT 220-5DNM analytical balance. It was considered that the obtained jelly met the quality conditions established in the pharmacopeia if its weight did not deviate by more than 7.5% from the average value. The analysis to determine the mass uniformity of the samples was performed in triplicate, and the obtained values were expressed as mean standard deviation (±SD).

To determine the production yield of the gelatin-based gummy jellies, we followed the methodology described by Najmuddin, M., et al. [24], with certain modifications. For this analysis, the total weight of the twenty gummy jellies (OJs) that we produced was measured using a Kern ABT 220-5DNM analytical balance to determine the practical mass. This practical mass was then divided by the theoretical mass, which represents the sum of all component masses that were used in the jelly formulation. The resulting ratio was multiplied by 100 to calculate the production yield, as expressed by the following Formula (1):(1)$$Yield\%=\frac{Mp}{Mt}\times 100$$
where Mp represents the practical mass, and Mt represents the theoretical mass. All determinations were made in triplicate, and the results were expressed as the mean value (±SD).

### 2.5. Determination of the Swelling Index

The analysis of the swelling index (SWL%) of the JO/EJO jellies was carried out by suspending one dry jelly of a known mass in 25 mL of five different pH media:

Simulated gastric fluid: This consisted of 0.1 M HCl, adjusted to a pH of 1.2.

Artificial saliva: Prepared using NaCl, KCl, KH_2_PO_4_, and NaHCO_3_ at pH levels of 5.0, 6.8, 7.3, and 8.0, which we prepared according to the provisions of the *European Pharmacopoeia 10.0* and the *Romanian Pharmacopoeia Ed.* and according to the provisions of the works published by Bechir, F., et al., and Pytko-Polonczyk, J. J., et al. [22,25,26,27].

The swelling index of the jellies was determined according to the method described by Frent et al. [28], to which some changes were made, taking into account Equation (2). This analysis was carried out for 60 min, and weighing was carried out on a Kern ABT 220-5DNM analytical balance until the samples reached a constant weight. The analysis to determine the swelling index of the samples was performed in triplicate, and the obtained values were expressed as the mean (±SD).(2)$$Swelling\;index\;(\%)=\frac{M_{ss}{-M}_{sd}}{M_{sd}}\times 100$$
where M*ss* represents the mass of the swollen samples, and M*sd* represents the mass of the dry samples.

#### Determining the Disaggregation Time

In order to determine the disaggregation time, we used an Electrolab TDT-08L (USP) device consisting of 8 glass tubs with a capacity of 1000 mL each and a thermostatic bath at a temperature of 37 ± 0.5 °C. Each tub was equipped with a vertical metal rod that rotated at 30 rpm in 300 mL of liquid medium from each tub in which the analyzed samples were introduced. The disaggregation test is considered appropriate when the residue in each tub does not present a palpable core, being made up of a soft mass.

In order to determine the disaggregation time of the analyzed samples, distilled water at different pH values, prepared according to the relevant formula, was used as a disaggregation medium.

### 2.6. Determination of the Tensile Strength

In the *Ph.Eur. 10th*, gummy gels are included in the group of oromucosal preparations as semi-solid preparations containing pharmacologically active ingredients that are intended for administration in the oral cavity and/or throat to achieve a local effect. As a result, to ensure mucoadhesion, they will be retained in the oral cavity by adhering to the mucosal epithelium so that they can release the substances [29]. The evaluation of the breaking strength characteristics of the gummy jellies was carried out by determining their tensile strength [30]. The equipment used was a TA.XTPlusC Texture Analyzer (Stable Micro Systems—Mason Technology Ltd., London, UK) equipped with a 5 kg load cell, with a 50 mm cylindrical plate and 2 compression cycles, using a speed of 5 mm/s test. All experiments were performed at room temperature, in compression mode, by triplicate testing of each gummy jelly, and the results were presented as mean ± standard deviation (SD).

### 2.7. In Vitro Release of Active Ingredients from Gummy Jellies and Evaluation of Functional Activity

In vitro diffusion was evaluated using a Franz cell system with synthetic membranes. The setup consisted of six diffusion cells, each with a diffusion surface area of 1.767 cm^2^ and a receptor compartment volume of 6.5 mL. The receptor chamber of each cell was filled with phosphate-buffered saline (pH 7.4), mixed with freshly prepared, heated, and degassed 30% ethanol.

The synthetic membranes were hydrated by immersion in the receptor medium for 30 min prior to use and then mounted between the donor and acceptor compartments of the Franz diffusion cell. A sample of approximately 0.100 g of jelly, enriched with oregano oil, was accurately weighed and applied to the membrane surface on the upper part of the diffusion cell.

The diffusion cells were tightly sealed by securing the dosing capsule with a clamp to prevent vehicle evaporation and to maintain the integrity of the samples throughout the experiment. The system was maintained at a constant temperature of 32 ± 1 °C, with continuous stirring at 600 rpm using a magnetic stirrer to minimize diffusion layer effects. Samples of 0.5 mL were extracted from the receptor solution at time intervals of 5, 10, 15, 30, 60, 120, 180, 240, 300, and 360 min. Each extracted volume was immediately replaced with fresh receptor medium to maintain a constant volume of 6.5 mL throughout the test.

The collected samples were analyzed using a UV-VIS spectrophotometer(Thermo Fisher Scientific GmbH, Dreieich, Germany) at 370 nm to determine the concentration of active ingredients that were released from the gummy jelly. Each sample was tested in triplicate to ensure the consistency of the results.

### 2.8. Composition Identification and Nutritional Quantification

The percentagewise composition of the oregano oil in the gel was verified using gas chromatography–mass spectrometry (GC-MS). A GC–MS analysis of 1% of oregano oil gel in coconut oil was conducted using a Thermo GC–MS (Trace 1310 ISQ 7000, Dreieich, Germany) with an HP−5MS capillary column (30 m × 0.32 mm × 0.25 µm). The electron ionization system operated at 70 eV with helium as the carrier gas at 30 cm/s. A 1 µL sample (1/100, *v*/*v* in dichloromethane) was injected in split mode (split ratio 120:1). The injector and mass transfer line temperatures were 290 °C and 220 °C, respectively. The oven temperature was programmed to increase from 45 °C (1 min) to 250 °C at 5 °C/min, staying at 250 °C for 5 min.

The relative percentages of compounds were determined by peak area normalization, and retention indices (Kovats) were used for qualitative and semi-quantitative analyses. These indices aid in identifying compounds, studying stationary phase properties, and analyzing thermodynamic relationships like vapor pressures and enthalpies. The results detailed in Table 2 show carvacrol as the major constituent, accounting for 15.22% of the extract.

The nutritional composition was calculated using the USDA Food Database and verified by standard laboratory analysis using gas chromatography for lipid content and Kjeldahl analysis for protein content.

The jelly composition table outlines the ingredients that were used in the formulations of the empty jelly (EOJ) and jelly with oil (OJ), highlighting their roles in the formulation (Table 3). The EOJ served as the base formulation, while the OJ included oregano oil as the active therapeutic agent. Both jellies contained similar ingredients, such as gelatin (thickening agent), glycerin (plasticizer), grape seed oil, date syrup (sweetener), citric acid (preservative), and water (solvent). The inclusion of oregano oil in the OJ added bioactive properties, such as antimicrobial and antioxidant effects.

Despite the Nutri-Score C (EJO) and D (JO) ratings, these functional food jellies combine a high protein content, low fat, and bioactive properties from natural ingredients like oregano oil. This positions them as healthier alternatives to standard jellies, appealing to consumers seeking natural, functional foods with reduced reliance on artificial additives.

### 2.9. Statistical Analysis

The data analysis was conducted using SPSS software, version 20 (IBM Corp., Boston, MA, USA). The results were presented as mean values with standard deviations. Statistical tests, including paired *t*-tests, ANOVA, Wilcoxon signed-rank tests, and Kruskal–Wallis tests, were applied based on the data distribution to evaluate changes and compare variables. Pearson correlation coefficients were calculated to assess relationships between variables. All analyses adhered to a significance level of *p* < 0.05, and data normality was verified before performing statistical tests. Missing data were addressed using a complete-case analysis approach.

## 3. Results

### 3.1. Mass Uniformity

The mass uniformity of the oregano-oil-enriched jellies (JO) met the *Romanian Pharmacopoeia 10th edition* criteria, with an average mass of 4.595 ± 0.147 g and a production yield of 91.9 ± 0.11%.

The 20 jellies that were loaded with oregano essential oil presented individual mass values ranging between 4.171 ± 0.037 g and 5.171 ± 0.095 g, with an average mass of 4.595 ± 0.147 g. The jelly production yield was 91.903 ± 0.113%. This indicates that the chosen method of preparation was effective, ensuring the uniform distribution of all components within the gummy mass. Gelatin functioned as a gelling agent, providing homogeneity to the preparation. Our results are similar to those of other authors who prepared jellies based on ethylephrine, using natural gelling agents like pectin, guar gum, tragacanth gum, xanthan gum, and sodium alginate. For example, Hassen Elshafee, A., and Moataz El-Dahmy, R., reported average jelly mass values ranging between 2.85 ± 0.15 g and 3.12 ± 0.23 g, which indicates that the mass distribution and uniformity of their jellies are consistent with ours [23].

### 3.2. Swelling Index

The results for water absorption by JO/EJO at different time intervals and pH values are shown in Figure 2.

The pH values at which the swelling index determinations were performed were 1.2, 5.0, 6.8, 7.3, and 8.0, and the time points were 10, 20, 30, 40, 50, and 60 min.

We subjected the gelatin-based gummy jellies, both with and without oregano oil in the composition, to the swelling index (SWL%) test to evaluate their ability to absorb water into their structure and to identify the artificial saliva medium to which they had the greatest affinity [31].

According to the results, it was observed that the blank samples without oregano oil (EJO) were much more hydrophilic and exhibited the highest SWL% values compared to those containing oregano oil (JO) across all analyzed pH media during the 60 min [23]. The presence of oregano oil in the jelly structure significantly influenced both the swelling index values and the swelling behavior of the jellies. Due to the intermolecular interactions between the gelatin structure and oregano oil, the gelatinous mass of the jelly became firmer, preventing a large amount of water from entering its structure [32].

In the artificial saliva medium with a pH of 6.8, after 60 min, the EJO exhibited the highest SWL% swelling index value (208.12 ± 0.35) at pH 6.8 after 60 min, while the JO in the artificial saliva medium with a pH of 5 showed the lowest SWL% value (128.76 ± 0.67). The jellies showed the highest swelling index values in alkaline artificial saliva media and the lowest in acidic media under similar conditions in the following order: pH 6.8 > pH 7 > pH 7.3 > pH 8 > pH 1.2 > pH 5. This suggests that the swelling index of the jellies was influenced by the composition of the preparation and the pH of the swelling medium. Gelatin, being a natural polymer with a three-dimensional structure, has the ability to swell in the presence of water due to the hydrophilic groups (-NH_2_ and -COOH) in its structure, which form new hydrogen bonds [31,33]. According to the literature, it is known that the pH value of the swelling medium affects the swelling behavior of gelatin in jellies [34].

In the artificial saliva medium with a pH of 6.8, the gelatin showed maximum swelling capacity. At this pH, the carboxyl (-COOH) groups in the gelatin structure are mostly deprotonated, which reduces the electrostatic repulsion between the gelatin chains. This allows for the formation of a stable network, promoting hydrogen bonding interactions with water molecules. Thus, gelatin absorbs water efficiently, leading to a significant expansion in volume.

In contrast, in the artificial saliva medium with a pH of 5, gelatin undergoes structural changes that affect its swelling capacity. At this acidic pH, the carboxyl groups can be protonated, which increases the electrostatic repulsion between the gelatin chains. This destabilizes the gelatin network, reducing its ability to form hydrogen bonds with water. As a result, gelatin absorbs much less water, which leads to reduced swelling and a low SWL% value.

### 3.3. Disintegration Time

The in vitro disintegration test is crucial for assessing the bioavailability of pharmaceutical forms, as the release of the active substance represents the initial step in the bioavailability process, which is important for bioequivalence studies [35,36]. This test helps identify optimal formulations and is widely used for quality control purposes. The disintegration time depends on factors such as the active substance, excipients, manufacturing process, and the disintegration environment.

Disintegration times were evaluated in artificial saliva at pH levels of 5.0, 6.8, 7.3, and 8.0, which represent typical conditions in the oral cavity [37,38,39], as well as in simulated gastric juice at a pH of 1.2 [40]. The fastest disintegration occurred at a pH of 6.8 (6 min at 37 °C and 30 rpm), followed by a pH of 7.3 (11 min), pH of 5 (12 min), and pH of 8 (17 min) [41,42,43]. The slower disintegration times at pH levels of 7.3 and pH 8 may be attributed to the alkaline conditions, which impact the hydration and swelling of the gelatin matrix [40,44,45,46].

Mechanical movement, simulated by agitation at 30 rpm, significantly reduced the disintegration time compared to tests without agitation. For instance, at a pH of 6.8, the disintegration time increased from 6 min (with agitation) to 13 min (without agitation). In distilled water, disintegration occurred more slowly than in artificial saliva under similar conditions, highlighting the influence of pH and dissolved ions on the gelatin matrix [47,48].

The presence of oregano oil also impacted the disintegration time by increasing the robustness of the gelatin matrix. In simulated gastric juice at a pH of 1.2, the disintegration time was similar to that observed in artificial saliva at a pH of 5, demonstrating the gelatin’s stability in acidic environments. Additionally, the pH of the disintegration medium was affected by oregano oil’s acidic nature, which may have implications for potential irritation in the oral cavity [49,50,51,52,53,54,55,56,57,58].

These results demonstrate the importance of pH and mechanical factors in the disintegration behavior of gelatin-based jellies, providing insights into their potential as functional delivery systems (Figure 3).

The disintegration tests revealed variations in disintegration times across different pH environments and under the influence of mechanical agitation. Oregano oil significantly prolonged the disintegration time in both distilled water and artificial saliva. In distilled water, disintegration occurred more slowly compared to in artificial saliva under identical conditions (composition, temperature, and paddle rotation), suggesting that the environmental pH plays a key role in the breakdown of the jellies. Even in the absence of oregano oil, faster disintegration was observed in distilled water compared to artificial saliva.

Mechanical agitation further influenced the disintegration process. The metal paddles accelerated disintegration, with disintegration times doubling in vats without paddles when distilled water was used as the medium. Similarly, in artificial saliva, longer disintegration times were recorded without agitation, irrespective of the pH level.

The disintegration tests of the oregano-oil-enriched jellies (JO) in artificial saliva at varying pH levels showed the fastest breakdown at a pH of 6.8, completing in 6 min at 37 °C with 30 rpm. At a pH of 7.3, disintegration occurred after 11 min, followed by 12 min at a pH of 5, and 17 min at a pH of 8. These differences were attributed to the pH of the disintegration medium, as all samples had identical compositions and underwent the same processing.

To simulate the effect of oral cavity dynamics, additional tests were conducted without paddles. Under these conditions, the disintegration times were longer, increasing to 13 min at a pH of 6.8, 15 min at a pH of 7.3, and 18 min at pH levels of both 5 and pH 8. These findings align with previous research, where reduced mechanical agitation resulted in longer disintegration times.

Given the occasional occurrence of highly acidic conditions in the oral cavity, tests at a pH of 1.2 in simulated gastric juice were performed. The results mirrored those in artificial saliva at a pH of 5, with disintegration times of 18 min under both static and agitated conditions, underscoring the robustness of the gelatin matrix in acidic environments.

The presence of oregano oil not only influenced the disintegration times but also altered the pH of the medium. Under both static and agitated conditions, jellies without oregano oil exhibited shorter disintegration times across all pH levels. The pH changes that were observed in the disintegration medium after the breakdown of the oregano-oil-enriched jellies (shown in Figure 4) highlight the impact of the oil’s acidic nature, which could have implications for potential oral cavity irritation.

The differences in the pH values of the disintegration medium after the disintegration of the jellies are due to the influence of the acidic pH of the oregano oil in the JO composition. Once released from the jellies, the oregano oil alters the pH of the disintegration medium. This is significant, because it provides insights into potential irritations that may occur in the oral cavity following the administration of these pharmaceutical forms, particularly in cases where the pH is altered from its normal value.

### 3.4. Tensile Strength

The measurements that were taken showed breaking strength values, measured in Newtons, indicating that the texture of the gummy jellies did not undergo significant changes with the addition of oregano essential oil. As shown in Figure 5, similar values for the tensile strength of the two types of jellies (JO/EJO) can be observed. Tensile strength measurements were expressed in N, with each result representing the mean ± SD. The number of replicates was *n* = 3 for each test [59].

No significant differences were detected between the two formulations, with tensile strength values of 5.2 ± 0.3 N for both the JO and EJO. This indicates that oregano essential oil does not influence the tensile strength of the gummy jellies. Compared to other studies, the tensile strength of oregano-oil-containing jellies shows values that demonstrate sufficient resistance [18].

## 4. Discussion

This study aimed to evaluate the effects of incorporating oregano oil, a natural extract known for its bioactive properties, into gelatin-based gummy jellies. The results were compared to findings from the existing literature to assess the impact on mass uniformity, swelling index, disintegration time, and tensile strength.

In terms of mass uniformity, the jellies exhibited a consistent distribution of oregano oil, with individual mass values within the acceptable ranges, as specified by the *Romanian Pharmacopoeia 10th edition*. These findings align with previous studies that also reported good mass uniformity when natural extracts were incorporated into gelatin matrices [60]. The results indicate that oregano oil did not disrupt the uniformity of the gummy structure, demonstrating that gelatin is a suitable carrier for bioactive compounds in functional foods.

The swelling index results highlight the significant impact of oregano oil on the water absorption behavior. The reduced swelling in acidic environments (pH 5) aligns with the hydrophobic interactions between oregano oil and the gelatin matrix, which reinforces the gel network [61]. These findings suggest a potential for controlled release applications, particularly in acidic gastric conditions [62]. However, the exclusion of pepsin in the gastric simulation is a limitation that warrants further investigation. Additionally, the disintegration times at varying pH levels indicate that the jellies are suitable for both oral and gastric environments, although modifications for more alkaline conditions could improve their versatility. This suggests that oregano oil could be valuable in controlling the release of active compounds, depending on the environment.

The disintegration time of the jellies depended on the pH and mechanical movement, with oregano oil prolonging it, particularly in acidic conditions (pH 5), aligning with research showing that essential oils slow the disintegration of gelatin-based products [63]. This delay in breakdown is beneficial for controlled release applications, enabling the gradual release of bioactive compounds in the digestive system. Compared to other studies focusing on natural extracts like pectin and guar gum, which exhibited faster disintegration due to their hydrophilic nature, our findings suggest that oregano oil can provide a more sustained release profile [64], particularly in acidic environments like the stomach.

The tensile strength of the jellies was not significantly affected by the incorporation of oregano oil. This finding aligns with previous research, such as Vojvodić Cebin et al., 2024 [18], where the addition of natural extracts did not alter the mechanical properties of gummy candies. In our study, the structural integrity of the jellies remained stable, ensuring durability during handling and consumption. This mechanical stability is essential for maintaining product quality and consumer acceptability, even when natural bioactive compounds like oregano oil are included.

In comparison to other studies that have explored the incorporation of natural extracts into functional foods, our study contributes valuable insights into the use of oregano oil specifically. While many studies focus on plant extracts or vitamins (e.g., Vitamin C, pectin) or role of vitamin D [65,66], the unique properties of oregano oil—particularly its impact on the swelling index and disintegration time—offer a novel approach to enhancing the functional properties of jellies [67]. Our results support the conclusions of studies like those by Oliveira, which emphasize the potential of gelatin-based products to act as carriers for hydrophobic bioactive compounds, while also providing unique insights into the specific behaviors of oregano oil.

While this study offers promising results, it has limitations. The long-term stability of oregano oil in the gelatin matrix under varying storage conditions (e.g., temperature, humidity) was not assessed, and further research is needed on its shelf life and compound degradation. The study primarily focused on physical properties, leaving bioavailability, therapeutic efficacy, and interactions with other ingredients unexplored. Additionally, while simulated pH environments were used, in vivo studies are necessary to understand the jellies’ behavior in real biological systems.

In summary, the incorporation of oregano oil into gelatin-based jellies resulted in a product with consistent mass uniformity, altered swelling properties, controlled disintegration time, and stable mechanical strength. Despite these promising results, further research is needed to explore the long-term stability and bioavailability of the jellies, as well as their behavior in biological systems. These findings contribute to the growing body of research on functional foods, highlighting the potential of oregano oil to enhance the health benefits of jellies without compromising their physical properties.

## 5. Conclusions

This study demonstrates the potential of oregano-oil-enriched gelatin-based jellies as functional food products with controlled release properties. The incorporation of oregano oil enhanced the disintegration time and reduced the swelling index without compromising the tensile strength. These findings support the application of such jellies for health-promoting uses. Future work should focus on long-term stability, bioavailability, and alternative plant-based matrices to address consumer demands for sustainable and allergen-free products.

## Figures and Tables

**Figure 1 foods-14-00479-f001:**
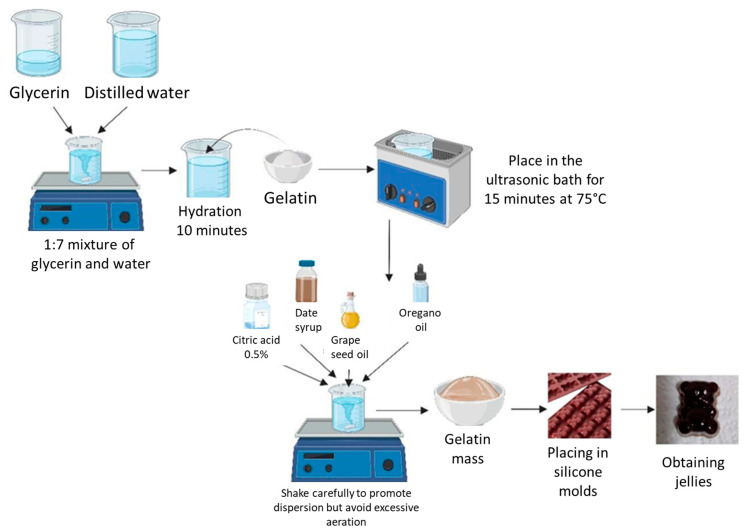
Schematic of preparation method of JO/EJO gummy jellies.

**Figure 2 foods-14-00479-f002:**
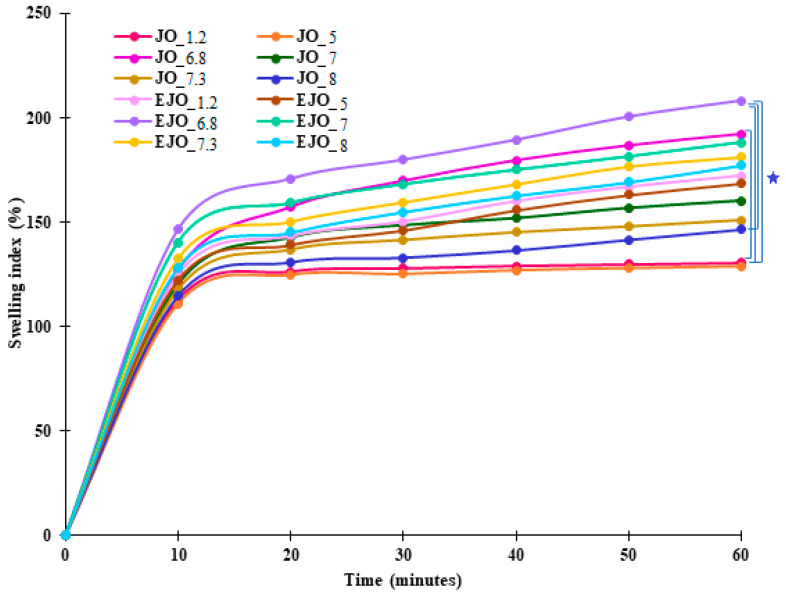
Swelling index of JO/EJO at different pH levels. 
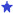
 = statistically significance (*p* < 0.05).

**Figure 3 foods-14-00479-f003:**
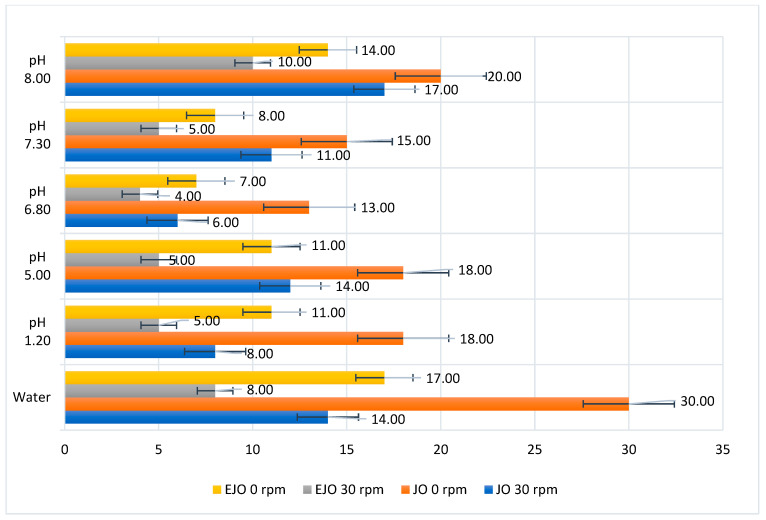
Disintegration times of JO/EJO at different pH levels.

**Figure 4 foods-14-00479-f004:**
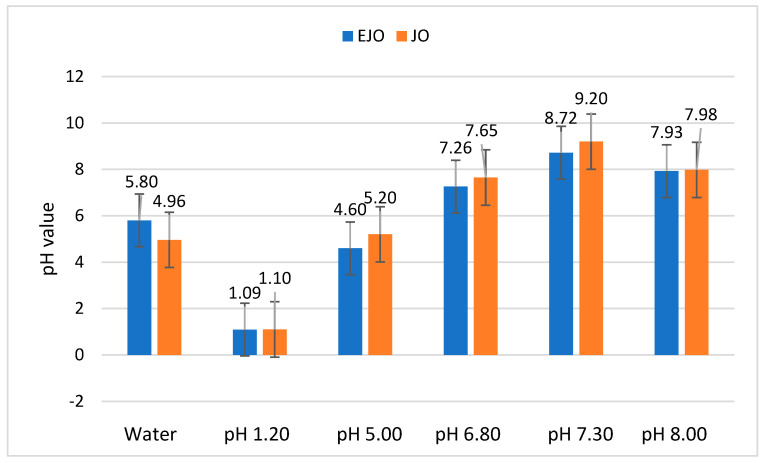
The pH values of the disintegration medium after the disintegration of EJO and JO at 30 rpm.

**Figure 5 foods-14-00479-f005:**
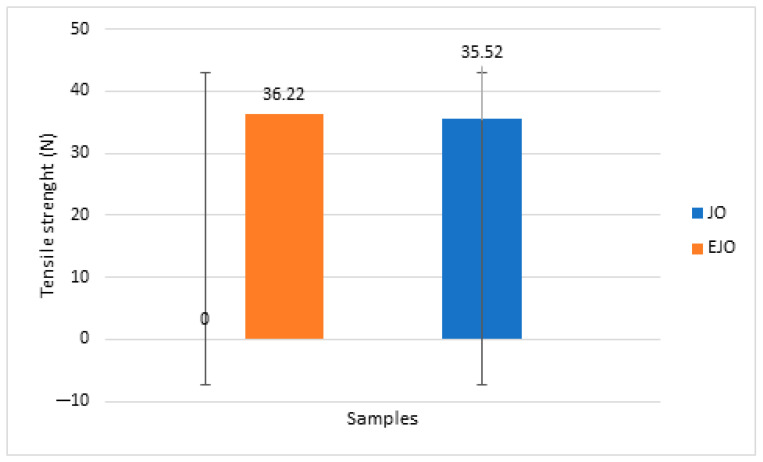
Tensile strength comparison between JO and EJO.

**Table 1 foods-14-00479-t001:** Formulas of gummy jellies with/without essential oil of oregano.

The Amount of Materials Used	The Amount of Materials Used						
	Oregano Oil (g)	Grape Seed Oil (g)	Date Syrup (g)	Citric Acid (g)	Gelatin (g)	Glycerin (g)	Distilled Water (g)
JO	0.5	5	20	0.5	20	10	q.s.
EJO	-	5	20	0.5	20	10	q.s.

JO = gummy jellies with essential oil of oregano; EJO = gummy jellies without essential oil of oregano, q.s. = as much as needed.

**Table 2 foods-14-00479-t002:** Gas chromatography report (g × 100 g^−1^ DW) of percentagewise composition for 100 g of dry extract of EOO.

Compound Name	Retention Time (Seconds)	Relative Area (%)	Kovats Retention Indices (RI)
Carvacrol	16.77	15.12	1292
endo-Borneol	9.737	3.39	1214
R-(−)-*p*-Menth-l-en-4-ol	9.890	2.89	1409
β-Bisabolene	14.349	2.65	2264
Linalool	11.92	2.02	1861
Caryophyllene	13.356	1.44	1962
Terpineol	10.210	1.06	1381
Caryophyllene oxide	15.397	1.01	3669
Carvone	11.118	0.61	1299
.tau.-Cadinol	15.989	0.57	1804
2,7-Dimethyloctadiin-3,5-diol-2,7	13.734	0.52	2288
Thymol	16.51	0.48	1268
(−)-Spatulenol	15.302	0.46	1829
3-Trifluoroacetate ester pregnenolone	21.107	0.46	3162
Eucaliptol	7.622	0.40	1670
Phenol, 2-methil-5-(1-methyletil)-, acetate	12.574	0.32	1953
*o*-lzopropilphenethol	12.033	0.27	1895
α-Pinene	6.074	0.23	713
Geranyl acetate	12.656	0.23	4360

**Table 3 foods-14-00479-t003:** Composition of and nutritional information for functional jellies (empty jelly and jelly with oregano oil).

Excipients	Empty Jelly (EOJ)	Jelly with Oil (OJ)	Role in Formulation	Nutritional Information (Per 100 g)
Oregano oil	-	30 drops (~1.5 g)	Therapeutic agent	9 kcal, 0 g protein, 0 g carbs, 1 g fat
Gelatin	20.00 g	20.00 g	Thickening agent	82 kcal, 20 g protein, 0 g carbs, 0 g fat
Glycerin	10.00 g	10.00 g	Plasticizer	40 kcal, 0 g protein, 10 g carbs, 0 g fat
Grape seed oil	5.00 g	5.00 g	Plasticizer	45 kcal, 0 g protein, 0 g carbs, 5 g fat
Date syrup	20.00 g	20.00 g	Sweetener	60 kcal, 0 g protein, 15 g carbs, 0 g fat
Citric acid	0.50 g	0.50 g	Preservative	0 kcal, 0 g protein, 0 g carbs, 0 g fat
Water	Ad 100.00 g	Ad 100.00 g	Solvent/Carrier	0 kcal, 0 g protein, 0 g carbs, 0 g fat
Calories	227 kcal	236 kcal	Reduced calories per unit	
Carbohydrates	25 g	25 g	Carbohydrates from natural origins	
Proteins	20 g	20 g	Proteins from natural sources	
Lipids	5 g	6 g	Lipids from vegetable sources	
Nutri-Score	C	D	Healthier-alternative functional food	

## Data Availability

All the data processed in this article are part of the research for a doctoral thesis, which is being archived in the esthetic medical office, where the interventions were performed.

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
