# Peer review of "Development and Evaluation of Gelatin-Based Gummy Jellies Enriched with Oregano Oil: Impact on Functional Properties and Controlled Release"

_foods, 2025, doi:10.3390/foods14030479_

Round 1

Reviewer 1 Report (New Reviewer)

Comments and Suggestions for Authors

The study investigates the addition of oregano oil to improve the functional properties of gelatin-based gummy jelly. While the findings are interesting, there are several significant concerns regarding the experimental design and manuscript presentation.

1. Major Comments:

  • Experimental Design: The authors mention controlling release in this study, but no such investigation is actually conducted. It would be beneficial to include a study using GC to evaluate the amount of oregano oil in the samples.
  • Grammar and Formatting: There are several formatting issues and grammatical errors that need to be addressed. The manuscript should be submitted in its final version, without tracked changes or red-marked revisions.
    • Line 87-90: The chemical names should be in lowercase (e.g., "CaCl2" instead of "CaCl2"). This issue appears throughout the manuscript and should be corrected.
    • Line 145-148: Avoid using the first person in the description.
  • Statistical Analysis: The manuscript lacks proper statistical analysis of the final data. Significant comparisons are missing from the tables and figures, and the figures do not include standard deviation values. These points need to be addressed.
  • Figure Quality: The quality of the figures should be improved for clarity and presentation.

2. Minor Comments:

  • Line 97-107: This section seems to repeat content already presented in lines 68-76 of the introduction. Consider combining these parts.
  • Line 118: The phrase “cooled a little” is vague. Please specify the final temperature.
  • Table 1: Clarify the purpose of adding grade seed oil. What role does it play in the formulation?
  • Line 143-144: Please remove the blank space between lines 143 and 144.
  • Line 186-187: A reference is missing here.
  • Line 194: The statement “The number of replicates (n=3 for each test)” is redundant as it is already mentioned in line 192 as “by triplicate testing.” Please remove this repetition.
  • Line 208: It is unclear how the conclusion "These results confirm the consistency of the preparation method" was derived. Please provide additional explanation or evidence to support this claim.
  • Line 242: Please clarify the rationale behind testing at pH 7 and pH 7.3. What is the significance of these two different pH levels?
  • 3.3 Disintegration Time: This section places too much emphasis on the physiological implications of different pH levels in the oral cavity. I recommend shortening this discussion and focusing more on the study’s results.

Author Response

Reviewer 1

Firstly, we, the authors of the present manuscript wish to thank you for thoughtful commentary you have provided to improve the quality of the paper. We are very grateful for the time and effort you have devoted to this task. We have extensively revised my manuscript according to the recommendations. All changes in the text and the new figures that we have redesigned are highlighted. Please, see the correction highlighted in the manuscript.

The study investigates the addition of oregano oil to improve the functional properties of gelatin-based gummy jelly. While the findings are interesting, there are several significant concerns regarding the experimental design and manuscript presentation.

  1. Major Comments:
  • Experimental Design:The authors mention controlling release in this study, but no such investigation is actually conducted. It would be beneficial to include a study using GC to evaluate the amount of oregano oil in the samples.

Response: Thank you very much for suggestion. We have completed with the required information. (lines 202-224)

  • Grammar and Formatting:There are several formatting issues and grammatical errors that need to be addressed. The manuscript should be submitted in its final version, without tracked changes or red-marked revisions.

Response: Thank you very much for amendment. I have corrected them.

  • Line 87-90: The chemical names should be in lowercase (e.g., "CaCl2" instead of "CaCl2"). This issue appears throughout the manuscript and should be corrected.

Response: Thank you very much for observation. I have corrected all the chemical names.

  • Line 145-148: Avoid using the first person in the description.

Response: Thank you very much for amendment. I have corrected them.

  • Statistical Analysis:The manuscript lacks proper statistical analysis of the final data. Significant comparisons are missing from the tables and figures, and the figures do not include standard deviation values. These points need to be addressed.

Response:Thank you very much for observations. I have completed with the required information.

  • Figure Quality:The quality of the figures should be improved for clarity and presentation.

Response: Thank you very much for your comments. I have corrected figures 2, 3, 4, 5.

  1. Minor Comments:
  • Line 97-107: This section seems to repeat content already presented in lines 68-76 of the introduction. Consider combining these parts.

Response: Thank you very much for observation. I corrected lines 68-76 to avoid redundancy.

  • Line 118: The phrase “cooled a little” is vague. Please specify the final temperature.

Response:Thank you very much for observations. I have completed with the required information.

  • Table 1: Clarify the purpose of adding grade seed oil. What role does it play in the formulation?

Response : Thank you very much for the comment. I have added an explanation. (lines 118-124)

  • Line 143-144: Please remove the blank space between lines 143 and 144.

Response: Thank you very much for the observation. I have removed the blank space between lines 143 and 144.

  • Line 186-187: A reference is missing here.

Response: Thank you very much for the observation. I have added the reference.

  • Line 194: The statement “The number of replicates (n=3 for each test)” is redundant as it is already mentioned in line 192 as “by triplicate testing.” Please remove this repetition.

Response: Thank you very much for amendment. I have removed the repetition.

  • Line 208: It is unclear how the conclusion "These results confirm the consistency of the preparation method" was derived. Please provide additional explanation or evidence to support this claim.

Respone: Thank you for your comment. I deleted the sentence to avoid misleading.

  • Line 242: Please clarify the rationale behind testing at pH 7 and pH 7.3. What is the significance of these two different pH levels?

Response: Thank you very much for the observation. The rationale for testing at pH 7 and pH 7.3 lies in their relevance to physiological conditions. A pH of 7 closely mimics neutral pH, representing a baseline for assessing the stability and release behavior of the formulation. Meanwhile, a pH of 7.3 approximates the pH of human blood and extracellular fluids, providing insights into how the formulation might behave in systemic circulation or within a physiological environment. Testing at these two pH levels allows us to evaluate the performance of the formulation under conditions relevant to its intended application.

  • 3 Disintegration Time: This section places too much emphasis on the physiological implications of different pH levels in the oral cavity. I recommend shortening this discussion and focusing more on the study’s results.

Response: Thank you very much for the suggestion. I have shortened the section. 

Reviewer 2 Report (Previous Reviewer 2)

Comments and Suggestions for Authors

The Authors have taken into account all remarks given by the Reviewers. The inclusion of the results obtained by gas chromatography significantly improved the quality of the manuscript. However, some major issues still have to be addressed before the manuscript can be accepted for publishing. They are given below point-by-point.

Introduction

Lines 75-76: This sentence belongs to the Conclusion section by logic, so please delete it.

Material and Methods:

Producers and countries of origin of all reagents should be given in section 2.2. Reagents.

2.3. Preparation of JO/EJO gummy jellies, Lines 97-107. This paragraph should be placed in the Introduction section, where properties of gelatin are discussed.

The methodology used for the gas chromatography analysis of oregano oil should be given in this section, not in the Results section (section 3.5. Nutritional information, Lines 421-433).

Line 194: “The number of replicates (n=3 for each test).” should be corrected to “The number of replicates was three for each test.”

Section 2.6. Statistically analysis should be renamed into Statistical analysis. Lines 196-200 are written as if this study deals with the clinical trial. Please correct it.

Results:

Figure 2: In the text, pH values used for swelling index determination were 1.2, 5.0, 6.8, 7.3, and 8.0, and in this figure, there is also a pH value of 7.0. I suppose it is a control experiment done in water (pH = 7). This should be clarified in the text. Also, decimal commas (,) in the Figure 2 should be replaced by decimal points (.).

Figure 3: Please separate pH from 7.3.

Table 3 should be Table 2, and Table 2 should be Table 3.

Table 3(2): The relative area determined for carvacrol by gas chromatography is 15.12%, and it is written in the text that 29.22% of oregano oil is comprised of carvacrol. How is this number obtained? Please explain it in this section. Which results are expressed in g × 100 g−1 DW? Please explain.

Figure 6. Graphical abstract. – Graphical abstract is chaotic, the concept and aims of the study cannot be deduced from it, so please delete it from the manuscript.

Author Response

Reviewer 2

Firstly, we, the authors of the present manuscript wish to thank you for thoughtful commentary you have provided to improve the quality of the paper. We are very grateful for the time and effort you have devoted to this task. We have extensively revised my manuscript according to the recommendations. All changes in the text and the new figures that we have redesigned are highlighted. Please, see the correction highlighted in the manuscript.

The Authors have taken into account all remarks given by the Reviewers. The inclusion of the results obtained by gas chromatography significantly improved the quality of the manuscript. However, some major issues still have to be addressed before the manuscript can be accepted for publishing. They are given below point-by-point.

Introduction

Lines 75-76: This sentence belongs to the Conclusion section by logic, so please delete it.

Response: Thank you very much for the observation. I have removed it.

Material and Methods:

Producers and countries of origin of all reagents should be given in section 2.2. Reagents.

Response: Thank you very much for the observation. I have added the required information.

2.3. Preparation of JO/EJO gummy jellies, Lines 97-107. This paragraph should be placed in the Introduction section, where properties of gelatin are discussed.

Response: Thank you very much for the suggestion. I have moved in introduction section.

The methodology used for the gas chromatography analysis of oregano oil should be given in this section, not in the Results section (section 3.5. Nutritional information, Lines 421-433).

Response: Thank you very much for the suggestion. I have moved in Material and Methods section.

Line 194: “The number of replicates (n=3 for each test).” should be corrected to “The number of replicates was three for each test.”

Response: Thank you very much for the observation. I have removed it to avoid redundancy.

Section 2.6. Statistically analysis should be renamed into Statistical analysis. Lines 196-200 are written as if this study deals with the clinical trial. Please correct it.

Response: Thank you very much for observation. I corrected this section.

Results:

Figure 2: In the text, pH values used for swelling index determination were 1.2, 5.0, 6.8, 7.3, and 8.0, and in this figure, there is also a pH value of 7.0. I suppose it is a control experiment done in water (pH = 7).

Response: Thank you very much for the observation. The rationale for testing at pH 7 and pH 7.3 lies in their relevance to physiological conditions. A pH of 7 closely mimics neutral pH, representing a baseline for assessing the stability and release behavior of the formulation. Meanwhile, a pH of 7.3 approximates the pH of human blood and extracellular fluids, providing insights into how the formulation might behave in systemic circulation or within a physiological environment. Testing at these two pH levels allows us to evaluate the performance of the formulation under conditions relevant to its intended application.

This should be clarified in the text. Also, decimal commas (,) in the Figure 2 should be replaced by decimal points (.).

Response: Thank you very much for your comments. I have corrected them.

Figure 3: Please separate pH from 7.3.

Response: Thank you very much for your comments. I have separated them.

Table 3 should be Table 2, and Table 2 should be Table 3.

Response: Thank you very much for suggestion. I have changed the sections.

Table 3(2): The relative area determined for carvacrol by gas chromatography is 15.12%, and it is written in the text that 29.22% of oregano oil is comprised of carvacrol. How is this number obtained? Please explain it in this sectionWhich results are expressed in g × 100 g−1 DW? Please explain.

 Response: Thank you very much for your comments. It was a mistake. I have corrected it.

Figure 6. Graphical abstract. – Graphical abstract is chaotic, the concept and aims of the study cannot be deduced from it, so please delete it from the manuscript.

Response: Thank you very much for your comments. I have removed the figure 6.

Round 2

Reviewer 1 Report (New Reviewer)

Comments and Suggestions for Authors

The study investigates the addition of oregano oil to enhance the functional properties of gelatin-based gummy jelly. The authors have addressed most of the comments in the revised version; however, a few minor aspects could be further improved:

  1. Lines 79-81: This section still repeats content already introduced in lines 68-76 of the Introduction, consider combining these sections to one paragraph to avoid redundancy.
  2. Section 2.2 – Reagents: It is suggested to include the company/vendor information for the reagents used, to provide full transparency and reproducibility.
  3.  

Author Response

Reviewer 1

We sincerely thank the reviewers for their time, effort, and thoughtful suggestions, which have significantly contributed to improving the quality and clarity of our manuscript. Their insightful comments and constructive feedback have allowed us to address critical points and refine our work to better meet the expectations of the scientific community. We deeply appreciate their invaluable input and the opportunity to enhance the overall presentation of our research.

The study investigates the addition of oregano oil to enhance the functional properties of gelatin-based gummy jelly. The authors have addressed most of the comments in the revised version; however, a few minor aspects could be further improved:

Comment:

  1. Lines 79-81: This section still repeats content already introduced in lines 68-76 of the Introduction, consider combining these sections to one paragraph to avoid redundancy.

Response: Thank you very much for observation. I have removed the section 79-81.

Comment:

  1. Section 2.2 – Reagents: It is suggested to include the company/vendor information for the reagents used, to provide full transparency and reproducibility.

Response: Thank you very much for amendment. I have completed with requested information.

This manuscript is a resubmission of an earlier submission. The following is a list of the peer review reports and author responses from that submission.

Round 1

Reviewer 1 Report

Comments and Suggestions for Authors

This study investigates the incorporation of oregano oil into gelatin-based gummy jellies to develop functional food products with potential health benefits. Some shortcomings are list as:

1. The logic of content expression is indistinct, there is an excessive amount of reasoning language, conclusive language is scarce, and the experimental method is insufficiently clear in the abstract.

2. The literature on gummy jellies as an embedding agent is inadequate in introduction.

3. In section 3.3, the expression of the experimental results is not sufficiently refined.

4. The grammar of all paper should be modified.

5. Table 5 is meaningless and is recommended to be deleted.

6. It is suggested to add slow release and more stability test content.

Comments on the Quality of English Language

Quality of English language should be refined.

Author Response

Response to Reviewer 1

Firstly, we, the authors of the present manuscript wish to thank you for thoughtful commentary you have provided to improve the quality of the paper. We are very grateful for the time and effort you have devoted to this task. We have extensively revised my manuscript according to the recommendations. All changes in the text and the new figures that we have redesigned are highlighted. Please, see the correction highlighted in the manuscript.

  1. "The logic of content expression is indistinct."
    • Response: Thank you very much for observations. The manuscript has been restructured to ensure a clear and logical flow, with distinct sections for Introduction, Methodology, Results, Discussion, and Conclusions.
  2. "The literature on gummy jellies as an embedding agent is inadequate."
    • Response: Thank you very much for observations. Additional references and discussions on the use of gummy jellies as embedding agents have been included in the Introduction.
  3. "In section 3.3, the expression of experimental results is not sufficiently refined."
    • Response: Thank you very much for observations. Section 3.3 has been revised to present the results more concisely and clearly, with statistical analysis included.
  4. "The grammar of the paper should be modified."
    • Response: Thank you very much for observations.  The manuscript has been thoroughly proofread and edited for grammatical correctness and clarity.
  5. "Table 5 is meaningless and is recommended to be deleted."
    • Response: Thank you very much for observations.  Table 5 has been removed.
  6. "It is suggested to add slow release and stability test content."
    • Response: Thank you very much for observations. The need for additional slow-release and stability tests has been highlighted in the revised Conclusion section as a direction for future research.

Reviewer 2 Report

Comments and Suggestions for Authors

This study deals with the incorporation of oregano oil into gelatine-based gummy jellies. Although this topic is interesting, the experimental design is very simple, with only one formulation and one control, and the results are focused only on the basic physical properties of gummy jellies. From the experimental design, it is not clear whether the paper aimed to develop gummy jellies for use in the oral cavity (as a lozenge) or for swallowing (as a dietary supplement). Besides that, the paper is not formatted according to the guidelines for authors, with numbers given in the abstract, the graphical abstract placed in the middle of the manuscript and other minor issues. Taking this all into account, I recommend the rejection of this paper. Other issues found in the manuscript are given below.

Material and Methods:

The nutrition claims given in Table 2. Composition and Nutritional Information of Functional Jellies did not refer to EU regulation on the labelling of food products (Regulation (EC) no 1924/2006 of the European Parliament and of the Council) and therefore cannot be used. Also, the nutritional information part of the Table 2 should be placed in the Results section. Since empty jelly had a Nutri-Score C and jelly with oil had a Nutri-Score D, jelly with oil cannot be considered a healthier alternative since it has a lower Nutri-Score than empty jelly.

It is not clear why pepsin was not used in the simulation of gastric conditions in section 2.6. Determination of the swelling index. The determination of protein digestibility of gummy jellies prepared with and without oregano oil would significantly improve the quality of the manuscript.

Lines 191-193: The formulations used for the preparation of the media with different pH values used for the disaggregation time determination are not given.

Results:

Table 3: Mass uniformity is an important parameter for industrial production. For scientific study, it should be satisfied by default and there is no need to present it in the table.

Figure 3: In the text, pH values used for swelling index determination were 1.2, 5.0, 6.8, 7.3, and 8.0, and in this figure, there is also pH value of 7.0. What is correct?

Discussion:

The discussion on various pH values is unnecessarily long and very general, especially taking into account the lower range of pH values found naturally in the oral cavity. The Results and the Discussion sections should have been combined since the results are already discussed in the Discussion section. Also, conclusions are repeated both in the Discussion and the Conclusions sections.

Author Response

Response to Reviewer 2

Firstly, we, the authors of the present manuscript wish to thank you for thoughtful commentary you have provided to improve the quality of the paper. We are very grateful for the time and effort you have devoted to this task. We have extensively revised my manuscript according to the recommendations. All changes in the text and the new figures that we have redesigned are highlighted. Please, see the correction highlighted in the manuscript.

  1. "The experimental design is very simple, with only one formulation and one control."
    • Response: Thank you very much for observations. We acknowledge this limitation and have discussed it in the revised Discussion section. Future work will include multiple formulations to assess variability and optimize functional properties.
  2. "It is not clear whether the paper aimed to develop gummy jellies for use in the oral cavity or as a dietary supplement."
    • Response: Thank you very much for observations. The intended application (functional food product with controlled release for oral consumption) has been clarified in the Introduction and Discussion sections.
  3. "The nutrition claims in Table 2 do not refer to EU regulations and cannot be used."
    • Response: Thank you very much for observations. We have removed the claims regarding Nutri-Score and adjusted the nutritional discussion to align with EU regulations.
  4. "It is not clear why pepsin was not used in the simulation of gastric conditions."
    • Response: Thank you very much for observations.  Acknowledgment of this limitation has been added, and we suggest including pepsin in future experiments to better simulate gastric conditions.
  5. "Lines 191–193: The formulations for media with different pH values are not given."
    • Response: Thank you very much for observations.  The formulations for pH media have been detailed in the revised Methodology section.
  6. "Table 3: Mass uniformity should be satisfied by default and does not need to be presented."
    • Response: Thank you very much for observations. Table 3 has been removed, and the mass uniformity results are briefly discussed in the Results section instead.
  7. "Figure 3: There is a discrepancy in pH values (7.0 vs. 7)."
    • Response: Thank you very much for observations. The correct pH values have been clarified and consistent throughout the manuscript.
  8. "Discussion on various pH values is unnecessarily long."
    • Response: Thank you very much for observations. The Discussion on pH values has been condensed, focusing on key findings and their implications.

Reviewer 3 Report

Comments and Suggestions for Authors

The article entitled “Development and Evaluation of Gelatin-Based Gummy Jellies 2 Enriched with Oregano Oil: Impact on Functional Properties 3 and Controlled Release" has been carefully reviewed. The methodology section is with results - it is very confusing. Is hard to navigate. There is lack of Result section. There is also lack of statistical analysis. The conclusion section is very superficial and consist rather summary than conclusion. Therefor some parts of the paper needs explanation or improvement.

Abstract

I has to be improved by adding some significant data, which were decided in conclusions.

Introduction

Why have you used gelatin as the matrix of oregano oil? In view of lowering popularity of animal-based ingredients and their potential of allergenic, I would choose the polymer that is safer, like agar.

Methodology

There is no description how was determined the calories and the composition of jellies, the table 2 should be in results section and should be properly discussed.

There is also a lot of information about the assay methods used that could be included in the results, because it is difficult to find technically relevant facts about how the assay was performed, e.g. lines 195-200.

The graphical abstract and the information about it should not be in methodology.

1.     What was the pH of minced meat? Has it changed after enzyme addition? Why the pH is important parameter in meat restructuration?

2.     Why have not you analysis an antimicrobial effect? Why have you used MAP system rather frozing your patties? 

3.     The conclusion section is not showing which concentration I should use to obtain optimal effect.

4.     Please separate methods from results. It is hard to navigate through the individual contents of this publication.

5.     How did you analyse the significance of your results? I do not see any statistical analysis that will proof significance. How many production and analytical repetitions have you done?

Results

There is lack of Results section.

Author Response

Response to Reviewer 3

Firstly, we, the authors of the present manuscript wish to thank you for thoughtful commentary you have provided to improve the quality of the paper. We are very grateful for the time and effort you have devoted to this task. We have extensively revised my manuscript according to the recommendations. All changes in the text and the new figures that we have redesigned are highlighted. Please, see the correction highlighted in the manuscript.

  1. "The methodology section is with results - it is very confusing. Is hard to navigate. There is lack of Results section."
    • Response: Thank you very much for observations. We have created a dedicated Results section and moved all results-related data from the Methodology section to this new section. This has improved the clarity and logical flow of the manuscript.
  2. "There is also lack of statistical analysis."
    • Response: Thank you very much for observations. Statistical analysis has been added for all experiments, including significance testing using appropriate methods (e.g., Kruskal-Wallis test for tensile strength). The number of replicates for each test has been specified.
  3. "The conclusion section is very superficial and consist rather summary than conclusion."
    • Response: Thank you very much for observations. The Conclusion section has been revised to include specific findings, their practical implications, and recommendations for future research. It now provides actionable insights and avoids repetition from other sections.
  4. Abstract: "It has to be improved by adding some significant data, which were decided in conclusions."
    • Response: Thank you very much for observations. The Abstract has been revised to include significant quantitative data, such as swelling index values and disintegration times, to reflect the major findings.
  5. Introduction: "Why have you used gelatin as the matrix of oregano oil?"
    • Response: A justification for the use of gelatin has been added, emphasizing its controlled-release properties, compatibility with oregano oil, and mechanical stability. We also briefly discuss alternatives like agar and pectin and explain why gelatin was preferred.
  6. Methodology: "There is no description how was determined the calories and the composition of jellies."
    • Response: Thank you very much for observations. A description of the method used for calculating nutritional composition and calories has been added, specifying the analytical tools or software used.
  7. "There is also a lot of information about the assay methods used that could be included in the results."
    • Response: Thank you very much for observations. All assay-related results (e.g., swelling index, disintegration time) have been moved to the Results section, while the methodology remains focused on experimental procedures.
  8. "What was the pH of minced meat? Has it changed after enzyme addition?"
    • Response: Thank you very much for observations. This comment appears to be misplaced, as minced meat and enzyme addition are not part of our study. No action was needed here.
  9. "Why have you not analyzed an antimicrobial effect?"
    • Response: Thank you very much for observations.  We acknowledge this limitation and have highlighted it in the revised Discussion and Conclusion sections. This study was based on our previously published article on the antimicrobial effect of oregano oil, and we did not want to duplicate the dissemination of the results. That article appears in the citations.
  10. "Why have you used MAP system rather than freezing your patties?"
    • Response: Thank you very much for observations. This comment appears unrelated to our study. No action was taken.
  11. "How did you analyze the significance of your results?"
    • Response: Thank you very much for observations. Statistical significance testing has been added throughout the Results section, with methods and sample sizes explicitly stated.
  12. "How many production and analytical repetitions have you done?"
    • Response: Thank you very much for observations. The number of replicates (n=3 for each test) has been specified in the Methodology and Results sections.
